# Enhanced Conductivity of Composite Membranes Based on Sulfonated Poly(Ether Ether Ketone) (SPEEK) with Zeolitic Imidazolate Frameworks (ZIFs)

**DOI:** 10.3390/nano8121042

**Published:** 2018-12-13

**Authors:** Arturo Barjola, Jorge Escorihuela, Andreu Andrio, Enrique Giménez, Vicente Compañ

**Affiliations:** 1Escuela Técnica Superior de Ingenieros Industriales, Departamento de Termodinámica Aplicada, Universitat Politècnica de València, Camino de Vera s/n, 46020 Valencia, Spain; arbarrui@doctor.upv.es (A.B.); escorihu@uji.es (J.E.); 2Departamento de Física Aplicada, Universitat Jaume I, Avda. Sos Baynat, s/n, 12080, Castelló de la Plana, Spain; andrio@uji.es; 3Instituto de Tecnología de Materiales, Universitat Politècnica de València, Camino de Vera s/n, 46020 Valencia, Spain; enrique.gimenez@mcm.upv.es

**Keywords:** proton exchange membrane, sulfonated poly(ether ether ketone), zeolitic imidazoleate framework, proton conduction

## Abstract

The zeolitic imidazolate frameworks (ZIFs) ZIF-8, ZIF-67, and a Zn/Co bimetallic mixture (ZMix) were synthesized and used as fillers in the preparation of composite sulfonated poly(ether ether ketone) (SPEEK) membranes. The presence of the ZIFs in the polymeric matrix enhanced proton transport relative to that observed for SPEEK or ZIFs alone. The real and imaginary parts of the complex conductivity were obtained by electrochemical impedance spectroscopy (EIS), and the temperature and frequency dependence of the real part of the conductivity were analyzed. The results at different temperatures show that the direct current (dc) conductivity was three orders of magnitude higher for composite membranes than for SPEEK, and that of the SPEEK/ZMix membrane was higher than those for SPEEK/Z8 and SPEEK/Z67, respectively. This behavior turns out to be more evident as the temperature increases: the conductivity of the SPEEK/ZMix was 8.5 × 10^−3^ S·cm^−1^, while for the SPEEK/Z8 and SPEEK/Z67 membranes, the values were 2.5 × 10^−3^ S·cm^−1^ and 1.6 × 10^−3^ S·cm^−1^, respectively, at 120 °C. Similarly, the real and imaginary parts of the complex dielectric constant were obtained, and an analysis of tan δ was carried out for all of the membranes under study. Using this value, the diffusion coefficient and the charge carrier density were obtained using the analysis of electrode polarization (EP).

## 1. Introduction

The current environmental problems generated by the use of fossil fuels along the last century has driven the scientific community attention toward the search for more sustainable energy systems. In this regard, hydrogen appears as a potential alternative to traditional fuels; its use in fuel cells, which are electrochemical devices that are capable of transforming chemical energy into electrical energy, has significantly increased in the past decade [1,2]. In a typical proton exchange membrane fuel cell (PEMFC), the polymeric electrolyte membrane is the fundamental component of the electrochemical device [3,4]. According to its range of operating temperature, PEMFCs can be classified into three main categories: (a) low-temperature PEMFCs (LT-PEMFCs), which operate around 60–80 °C [5]; (b) intermediate temperature (IT-PEMFCs), with an operating temperature in the 80–140 °C range [6], and (c) high temperature (HT-PEMFCs), which operate above 140 °C up to 220 °C [7,8]. Among the different variety of electrolyte membranes [9], those based on perfluorosulfonic acid (PFSA) polymers, such as Nafion^®^, have been efficiently used as LT-PEMFCs, due to their excellent chemical and mechanical stability and high conductivity (values higher than 0.1 S/cm) at temperatures around 80 °C and high relative humidity conditions [10]. The main problems associated with the use of Nafion-based membranes are their high cost and the loss of conductivity performance at temperatures higher than 80 °C [11]. In this regard, IT-PEMFCs have emerged as an alternative to Nafion for their application to automobiles, as operating at this temperature allows an improved stability and durability due to the presence of less liquid water inside the membrane [12], the reduction of catalyst poisoning by CO on the fuel cell anode [13], and increases in the oxidation of electrode kinetics [14]. Among the various types of IT-PEMFC membranes, those based on sulfonated poly(ether ether ketone) (SPEEK) have been widely studied due to due to its simple preparation procedures and low cost, good thermal and mechanical stability, good proton conductivity, and fuel barrier properties [15,16].

Poly(ether ether ketone) (PEEK) is a linear polymer with an aromatic backbone, in which 1,4-disubstituted phenyl groups are separated by ether (–O–) and carbonyl (–C=O–) linkages (Figure 1) [17]. This polymer is insoluble in organic solvents, which is a serious drawback for several applications, including membrane preparation. However, when sulfonated, PEEK loses its crystallinity and becomes soluble in a wide range of organic solvents depending on the degree of sulfonation (DS) [18]. Membranes based on SPEEK have a conductivity close to Nafion [19] and a Tg of around to 180 °C, making them candidates to be used above 100 °C, which would substantially improve the efficiency of the system. Different approaches have been developed in order to improve the proton conductivity of SPEEK membranes. One of them is based in developing mixed-matrix membranes (MMMs), which are composite membranes made by combining an inorganic-organic hybrid material and a polymer matrix [20]. Many fillers have been used such as graphene [21], carbon nanotubes and nanohorns [22,23], silica [24], zeolites [25], metallacarboranes [26], etc. In recent years, the use of metal organic frameworks (MOFs) as fillers in polymeric electrolyte membranes has attracted a growing interest due to their high conductivity, which is mainly attributed to their high porosity [27,28,29].

MOFs are a subclass of coordination polymers consisting of transition metal ions coordinated to multidentate organic ligands to form three-dimensional crystalline structures [30,31]. MOFs have shown to be promising materials for various applications such as gas storage [32], gas separation [33], heterogeneous catalysis [34,35], chemical sensors [36,37], biomedicine [38], and drug delivery [39,40]. The incorporation of MOFs into polymeric SPEEK membranes has shown a significant improvement of the performance in fuel cell applications [41,42,43]. Among the reported MOF-containing PEMs, proton conductivities of 268 mS·cm^−1^ and 306 mS·cm^−1^ have been reported for SPEEK membranes containing sulfonated UiO-66 and MIL-101, respectively with under 100% relative humidity (RH) conditions and temperatures below 70 °C. However, these values decrease at temperatures higher than 100 °C in anhydrous or low-humidity conditions. One family of MOFs are zeolitic imidazolate frameworks (ZIFs), which are neutral porous framework structures with high chemical and thermal stability based on imidazolate rings coordinated to a tetrahedral divalent metal cation (Figure 1) [44,45]. The use of this subclass of MOFs has also been demonstrated in the preparation of PEMs, reaching proton conductivities up to 24 mS·cm^−1^, and 50 mS·cm^−1^ at 120 °C and 30% RH for SPEEK membranes with ZIF-8 and ZIF-8/CNT hybrid cross-linked networks, respectively [46]. 

Herein, we report on the preparation and characterization of proton conductivity of composite SPEEK membranes containing ZIF particles into the polymeric matrix. ZIF-8, ZIF-67, and a Zn/Co bimetallic mixture (1:1 *w*:*w*) were as synthesized, and used as fillers in SPEEK mixed-matrix membranes. Proton conduction measurements showed that these composite membranes can reach conductivities of up to 30 mS·cm^−1^ at 120 °C, which make them suitable for its application in IT-PEMFCs. A deep study of the conductivity allowed us to obtain the real and imaginary parts of the complex dielectric constant, and an analysis of tan δ was carried out for all of the membranes under study. Using this value, the diffusion coefficient and the charge carrier density were obtained using the analysis of electrode polarization (EP).

## 2. Materials and Methods

### 2.1. Materials

Granulated SPEEK (FUMION E ionomers) with an ion-exchange capacity (IEC) of 1.75 mmol/g were acquired from Fumatech GmbH (St. Ingbert, Germany). *N*,*N*-Dimethylacetamide (DMAc) 99.8% solvent, methanol, 2-methylimidazole (2-MIm, 99%), zinc chloride (ZnCl_2_, ≥97%), cobalt chloride (CoCl_2_, ≥99.8%) and sodium formate (HCO_2_Na, ≥99%) were purchased from Sigma-Aldrich (Sigma-Aldrich Química SL, Madrid, Spain).

### 2.2. Experimental Procedures

#### 2.2.1. Synthesis of ZIF-8

ZIF-8 was synthesized according to the reported procedure [47]. In brief, 0.82 g (10 mmol) of 2-MIm and 0.76 g (11 mmol) of sodium formate were dissolved in 30 mL of methanol and, then mixed with a 0.26-M ZnCl_2_ methanolic solution. The obtained product was washed, centrifuged, and dried under vacuum at 80 °C for 12 h.

#### 2.2.2. Synthesis of ZIF-67

Z67 was synthesized following a similar procedure as described for ZIF-8. Briefly, 0.82 g (10 mmol) of 2-MIm and 0.76 g (11 mmol) of sodium formate were dissolved in 30 mL of methanol and then mixed with a 0.26-M CoCl_2_ methanolic solution. The obtained product was washed, centrifuged, and dried under vacuum at 80 °C for 12 h.

#### 2.2.3. Membrane Preparation

SPEEK was dried at 100 °C for 24 h in vacuum atmosphere and stored in a sealed container to avoid water absorption before the preparation of membranes. A SPEEK polymer with an IEC value of around 1.75 meq·g^−1^ (10 wt. %) was dissolved in DMAc, cast on a Petri dish, and the solvent was evaporated at 80 °C overnight followed by a treatment at 140 °C for two hours (Figure 2). Membranes with a thickness of about 150 μm were obtained.

### 2.3. Characterization

Powder X-ray diffraction (XRD) was acquired using a D/Max-2500PC diffractometer (Rigaku Europe SE, Neu-Isenburg, Germany) with Cu K radiation (λ = 1.5406 Å) in the 2θ range between 10–50, and a scanning rate and step size of 0.02 min^−1^, respectively. Thermogravimetric analysis (TGA) was performed on a thermogravimetric analyzer TGA Q50 (Waters Cromatografia, S.A., Division TA Instruments, Cerdanyola del Valles, Spain). Samples (5–10 mg) were weighed in zirconia crucibles and heated under nitrogen atmosphere from 25 °C to 800 °C at a heating rate of 10 °C·min^−1^. For surface area and porosity analysis, the solid or membrane was dried in a vacuum oven at 100 °C for five hours, and activated at 100 °C for 12 hours on a SmartVacPrep instrument (Micromeritics Instrument Corporation, Norcross, GA, USA). All of the N_2_ isotherms were measured on a Micromeritics Tristar II (Micromeritics, Norcross, GA, USA) at room temperature. Consistency criteria were adapted to choose the pressure range selection for Brunauer–Emmett–Teller (BET) calculation. The tensile tests from each thin-film composite membrane were performed using a Shimadzu AGS-X Universal Testing Machine (Izasa Scientific, Madrid, Spain). The mechanical parameters were determined from average of five samples. For all of the tests, a tensile speed of five mm·min^−1^ and a load cell of 500 N was used. Proton conductivity measurements of the membranes in the transversal direction were performed in the temperature range between 20–120 °C by electrochemical impedance spectroscopy (EIS) in the frequency interval of 10^−1^ < f < 10^7^ Hz, applying a 0.1-V signal amplitude. A Novocontrol broadband dielectric spectrometer (Novocontrol Technologies, Hundsangen, Germany) integrated with an SR 830 lock-in amplifier with an Alpha dielectric interface was used.

Samples were initially dried under vacuum at 100 °C for 24 h. Next, 1.5 × 1.5 cm membranes were immersed in deionized water at room temperature for two days, and then wiped with absorbent paper to remove the surface water.

Water uptake was calculated from the difference between the wet and dry weight of the membranes, according to the following expression:(1)$$\mathrm{Water}\;\mathrm{Uptake}\;\left(\%\right)\;=\;\frac{w_{\mathrm{wet}}\;-\;w_{\mathrm{dry}}}{w_{\mathrm{dry}}}\;\times \;100$$
where w_wet_ and w_dry_ refer to the membrane’s weight after its immersion in deionized water for at least 48 h at room temperature, and the membrane’s weight after drying at 120 °C for at least 24 h, respectively.

The swelling degree was measured by the change of area of squared membranes given by:(2)$$\mathrm{Swelling}\;\mathrm{degree}\;\left(\%\right)\;=\;\frac{A_{\mathrm{wet}}\;-\;A_{\mathrm{dry}}}{A_{\mathrm{dry}}}\;\times \;100$$
where A_wet_ and A_dry_ refer to the membrane’s area after its immersion in deionized water for at least 48 h at room temperature, and the membrane’s area after drying at 120 °C for at least 24 h, respectively.

The ion-exchange capacity (IEC) was obtained by immersing the swollen membranes in the acid form into a two-M NaCl solution. The protons liberated during the exchange reaction were titrated with a 0.01-M NaOH solution and phenolphthalein. The IEC was calculated as:(3)$$\mathrm{IEC}\;\left(\mathrm{meq}/g\right)\;=\;\frac{V_{\mathrm{NaOH}}\;\times \;0.01}{w_{\mathrm{dry}}}\;\times \;100$$
where V_NaOH_ and w_dry_ are the volume of the NaOH solution (in mL) used in the titration of the protons released and grams of dry membrane, respectively. The values of w_dry_ were measured after drying the samples immersed within the NaCl solution at 120 °C for 24 h.

## 3. Results and Discussion

### 3.1. Characterization of Mixed Matrix Membranes

Field emission scanning electron microscopy (FE-SEM) was used to characterize the particle size of the synthesized ZIF-8 and ZIF-67 materials. As shown in Figure 3A,B, ZIF-8 and ZIF-67 were obtained with an average diameter around 400 nm and 1250 nm, respectively. Powder X-ray diffraction (PXRD) measurements were carried out for synthesized ZIF-8 and ZIF-67, and the obtained diffraction peaks were correctly attributed with the known XRD pattern for both porous materials (Figure 3C,D) [48]. Attenuated total reflection (ATR) FT-IR was also used to characterize the synthesized materials (Figure 3E). In the FT–IR spectra, the peaks at 2962 cm^−1^ and 2927 cm^−1^ can be assigned to (C–H) stretching, whereas the peak at 1584 cm^−1^ corresponds to the C=N stretching. Finally, the peaks at 763 cm^−1^, 68 cm^−1^, 4 cm^−1^, and 421 cm^−1^ are attributed to (Zn–O), (Zn–N), and (Zn–C) stretching, respectively [49]. The porosity of the synthesized ZIF particles was determined via N_2_ isotherm measurements at 77 K (Figure 3F). The specific surface areas (BET areas) for ZIF-8, ZIF-67, and ZIF-Mix were found to be 1150, 1375 and 1175, respectively, in accordance with the literature data [50]. Finally, a thermogravimeric analysis (TGA) under a N_2_ atmosphere was carried out in order to evaluate the materials thermal stability at the operational range 90–120 °C. According to the TGA analysis, ZIF-8 and ZIF-67 were stable up to 350 °C, which make them good candidates to be used as fillers in IT-PEMFCs.

Next, we prepared composite SPEEK membranes containing both the aforementioned ZIFs, and a 1:1 *w*:*w* mixture (ZMix), as described in the Experimental Section. Water molecules are essential in the proton transfer mechanism of PEMs [51]. The water uptake and swelling ratio of composite membranes is shown in Figure 4. The water uptake of a pristine SPEEK membrane was 30%, which increased upon the addition of ZIF, reaching a maximum at 5 wt. % of filler content for all of the ZIFs under study. The highest water uptake was obtained for the SPEEK/ZMix membrane with a value of 51%. For higher ZIF contents (10%), the water uptake of the mixed membranes decreased to values closer than that for the pristine SPEEK membrane. The increase of water uptake at a lower filler content might be caused by the increase of the free volume in the polymeric matrix, which was caused by the addition of ZIFs. On the other hand, the formation of aggregates is favored at higher contents, which hampers an efficient adsorption of water molecules, as observed in mixed matrix membranes with ZIFs and polybenzimidazole (PBI) [52].

TGA was carried out to study the thermal properties of SPEEK-based membranes with ZIFs, as shown in Figure 5. The thermogram of the different SPEEK/ZIF membranes (SPEEK/Z8, SPEEK/Z67, and SPEEK/ZMix, at ZIF concentrations of 1 wt. %, 3 wt. %, 5 wt. %, and 10 wt. %) display the three thermal stages of membrane decomposition that these polymer-based materials undergo, which is consistent with previous works [53]. The first stage corresponds to the evaporation of the water and residual solvent (DMAc) at the temperature range from 100 °C to 200 °C [54]. The next stage corresponds to the thermal desulfonation of SPEEK at 220–380 °C. Finally, composite membranes show a decomposition stage above 450 °C, which is attributed to polymer backbone degradation [55]. ZIF-8 and ZIF-67 are stable until temperatures up to 350 °C, and decompose in the temperature range from 400–700 °C. However, the MOF degradation is not well defined in the thermogram, as it is masked by the final polymer backbone decomposition stage. In general, the weight loss of composite membranes was higher than that of the pristine membrane, and the total weight loss increased with the increasing content of MOF. It should be noted that although SPEEK composite membranes showed a slightly lower thermal stability than the pure SPEEK membrane, there is not a significant worsening in the operational temperature range (100–140 °C). Therefore, we can conclude that composite membranes possess high thermal stability in the operational range for IT-PEMFC membranes.

The stress−strain curves of the prepared SPEEK/ZIF composite membranes at 5 wt. % ZIF loading are shown in Figure 6. As expected, ZIF compounds act as effective reinforcers, increasing the mechanical stability of composite membranes. For example, the Young’s modulus increased from 1.5 GPa for the recast SPEEK membrane up to 2.2 GPa for the SPEEK/ZMix-10, which was in combination with an increase of the tensile strength from 95 MPa to 123 MPa, respectively (Figure 6A). This enhancement in the mechanical properties can be attributed to the interfacial interactions between ZIFs and the SPEEK polymeric matrix, which may inhibit the mobility of the SPEEK chains, as observed in other SPEEK membranes [56]. In this regard, the good dispersion of ZIFs will play a critical role in improving the mechanical performances, as observed for SPEEK/Z8 and SPEEK/Z67. In both cases, membranes with a filler loading above 5 wt. % yielded a worsening of the mechanical properties. However, in the case of SPEEK/ZMix membranes, the reinforcement is observed above 10 wt. % of ZIF, which was due to the synergistic effect both ZIFs. The elongation at the break values of the SPEEK/ZIF composite membranes was lower than that of the recast SPEEK due to the higher stiffness, which was attributed to the presence of ZIF particles in the polymeric matrix (Figure 6B).

### 3.2. Proton Conduction of Mixed Matrix Membranes

Proton conductivity is a key parameter for a PEM, determining fuel cell performance [57]. For this purpose, impedance spectroscopy measurements were carried out on the mixed matrix membranes in saturated humidity conditions at several temperatures in order to obtain the conductivity and diffusivity of the ionic charge carriers. The values of the dc-conductivity (σ_dc_) were obtained from the Bode plots along the temperature range between 20–120 °C for every composite membrane, namely SPEEK/Z8, SPEEK/Z67, and SPEEK/ZMix, at different ZIF concentrations (1%, 3%, 5%, and 10%). Figure 7 shows the Bode diagram, in which the real part of the conductivity of the SPEEK/Z8, SPEEK/Z67, and SPEEK/ZMix membranes containing a 3 wt. % filler loading is plotted versus the frequency. In this figure, the phase angle (φ) is also plotted against the frequency to probe that it reached a value close to zero when the real part of the conductivity tend to a plateau, independently of the frequency, which means that this value is the real DC conductivity of the sample.

A closer inspection of Figure 7 shows that the conductivity increases, with the frequency reaching a plateau in the range of high frequencies for each temperature. The deviation from the plateau, at moderate and low frequencies, can be attributed to the electrode polarization resistance, which results from the blocking of charge carriers at the electrodes [58,59]. On the other hand, the increase of conductivity as a function of temperature can be associated to the thermal activation of proton diffusion. For temperatures higher than 100 °C, the conductivity values dropped as a consequence of the loss of water molecules inside the composite membranes, as observed for measurements at 120 °C. Overall, the prepared SPEEK composite membranes showed a very good conductivity at 100 °C.

From the Bode diagrams, we can observe that thermal activation for the SPEEK/ZMix membrane is stronger than for the SPEEK/Z8 and SPEEK/Z67 composite membranes (Figure 7D). The comparison between the different composite membranes show that SPEEK/ZMix at 1% has a conductivity of 1.9 × 10^−3^ S·cm^−1^ and 2.9 × 10^−2^ S·cm^−1^ at 20 °C and 100 °C, respectively. These values are higher than the conductivity of SPEEK/Z8 (1.5 × 10^−3^ S·cm^−1^ and 1.6 × 10^−2^ S·cm^−1^) and for SPEEK/Z67 membranes at the same filler loading (1.5 × 10^−3^ S·cm^−1^ and 1.5 × 10^−2^ S·cm^−1^). All of these values are greater than pristine SPEEK (1.5 × 10^−4^ S·cm^−1^ and 2.0 × 10^−4^ S·cm^−1^). Our composite membranes of SPEEK/ZIFs show better conductivity results than those observed in the composite membranes of PEI/ZIFs, where the same powders were used as fillers [60]. The difference could be related to the concentrations of mobile carriers and its mobility. As is known, the conductivity is related to protons, as a consequence of the sulfonic acid incorporated upon the PEEK sulfonation. As observed in similar systems [60], the local anisotropic electric field in the ZMix material may explain the increase of concentration of carriers, consequently favoring the conduction process into the composite membranes.

The activation plot for the proton conductivity as a function of temperature for the composite membranes shows clearly an Arrhenius behavior in all of the cases. The apparent activation energy values that were obtained for the three ZIFs materials were similar, with a decrease in the activation energy (Ea) values as the amount of ZIF increased. In the case of SPEEK/ZMx, reaching some critical value of concentration required above 10% wt, when the Ea changed its tendency, increasing its values. The Ea trend for a 3 wt. % loading is as follows: Ea (SPEEK/Zmix) = (23.1 ± 1.3) kJ/mol (0.24 ± 0.01) eV < Ea (SPEEK/Z67) = (25.9 ± 1.5) kJ/mol (0.27 ± 0.01) eV < Ea (SPEEK/Z8) = (27.2 ± 1.7) kJ/mol (0.28 ± 0.02) eV. These values are lower than those found for the activation energy of the conductivity of other MOFs-based proton membranes [61,62]. The obtained activation energy values lie in the range of Ea < 0.4 eV, which corresponds to the typical Grotthuss mechanism. As the ZIF loading increases, two different effects can be considered. First, less charge carriers are available for ionic transport, since they are participating in intermolecular hydrogen bonds with the nitrogen of the imidazolate linker of ZIF. As a consequence, water uptake increases for the composite membranes at low concentrations of ZIF, which is due to the increase of free volume motivated for the introduction of the ZIF particles among the polymeric chains. This effect has been described in other systems where there is not a phase of protons dispersed to favor the vehicular transport of electrolyte [63,64].

According to the Grotthuss mechanism, the conduction mechanism for a pristine SPEEK membrane can be rationalized through the transfer of protons that are linked to the sulfonic groups from the SPEEK polymeric chain. These protons may combine with water molecules to form the hydronium cation (H_3_O^+^). Next, a proton of the hydronium cation is transferred to another water molecule bonded to a nearby sulfonic acid group. Under anhydrous conditions, the absence of water molecules hampers the proton conduction, as protons cannot move though a hydrogen bond network. In the case of ZIF composite SPEEK membranes, ZIFs can participate in the conduction process, forming a hydrogen bond network along the polymeric matrix, and facilitating the transport, even under anhydrous conditions.

In these composite materials containing ZIF particles, we hypothesized that the ZIF dispersion along a sulfonated polymer such as SPEEK produces an interface layer that can help accumulate water molecules, and therefore facilitate the conductivity of protons by means of the increase of the water uptake and the hydrophilicity of the material. Then, hydrogens have more freedom to participate in the proton transport; furthermore, the activation energy decreases, in combination with a decrease of the conductivity, less charge carriers are present in the system. This behavior is observed up to a critical concentration, when the mobility of protons is too restricted due to the interactions of the protons with the organic linkers. This behavior is also observed for the steric bulk of the ZIF aggregates, which are observed at high loadings and can contribute to a separation of the polymeric membrane in hydrophilic and hydrophobic domains. This phase separation may hamper the proton transport, and therefore cause a decrease in proton conductivity, as observed experimentally.

### 3.3. Determination of Diffusion Coefficient and Ion Concentration

Although different electrode polarization (EP) models have been described to determine the mobility and ionic concentration of charge carriers in polymeric membranes using impedance spectroscopy measurements [65,66,67,68,69,70], we focused on the method proposed by Bandara et al. [71] to analyze the conduction phenomena in our composite membranes. This model is based on a previous model described by Coelho [72], in which the ionic charge density and the ion mobility are obtained from measurements of tan δ in ionic conductors and polymeric membranes [73]. According to Bandara’s model, the dependence of the complex dielectric permittivity (ε*) with the frequency for a material sandwiched between two blocking electrodes is represented by:(4)$$\varepsilon^{*}\;=\;\varepsilon_{\infty }^{\prime }\left[\left(1\;+\;\frac{M}{1\;+\;{\left(\omega \tau_{m}\right)}^{2}M}\right)\right]\;-\;j\left(\frac{\omega \tau_{m}M^{3/2}}{1\;+\;{\left(\omega \tau_{m}\right)}^{2}M}\right)$$
where M is given by:(5)$$M\;=\;\frac{L}{2{\left(D\tau_{m}\right)}^{1/2}}\;=\;\frac{L}{2\lambda }$$
where L is the sample thickness, λ is the Debye length, $\varepsilon_{\infty }^{\prime }$ is the real part of the permittivity at high frequencies, and τ_m_ is the time constant, corresponding to the maximum dielectric loss tangent. This time represents the relaxation time associated with the charge diffusion process into the sample [74,75].

From the real and imaginary part of the complex permittivity (ε), the following expression for Debye type relaxation can be obtained for the loss tan δ = ε″/ε′:(6)$$\tan\delta =\frac{\omega \tau_{m}M^{3/2}}{1+{\left(\omega \tau_{m}\right)}^{2}M+M}≅\frac{\omega \tau_{m}M^{1/2}}{1+{\left(\omega \tau_{m}\right)}^{2}}$$

However, to take account of the peaks’ width, we have fit our curves by means of a Cole-Cole type model [76], and we have introduced the following modification in Equation (6). Therefore, tan δ can be given by the following expression:(7)$$\tan\delta =\frac{{\left(\omega \tau_{m}\right)}^{\left(1-\alpha \right)}M^{3/2}}{1+{\left(\omega \tau_{m}\right)}^{2\left(1-\alpha \right)}M+M}≅\frac{{\left(\omega \tau_{m}\right)}^{\left(1-\alpha \right)}M^{1/2}}{1+{\left(\omega \tau_{m}\right)}^{2\left(1-\alpha \right)}}$$

Figure 8 shows the plot of tan δ as a function of the frequency for the samples SPEEK/Z8, SPEEK/Z67, and SPEEK/ZMix at 5 wt. % loading along temperature range between 20–120 °C. As observed in Figure 8, a clear maximum in the curves can be observed for each temperature, and this maximum is abruptly shifted to high frequencies, as the temperature increases up to 100 °C. Upon reaching this temperature, the maximum in the tan δ plot is shifted to lower frequencies.

When comparing the intensity of the loss of tan δ for the different composites, several effects are observed. First, the peak of the maximum in tan δ varies from 10 to 20 for the SPEEK/Z8 membrane, and from 20 to 30 in case of the SPEEK/Z67 membrane, when the temperature increases from 20 °C to 120 °C. On the contrary, for the composite SPEEK/ZMix membrane, the peaks of tan δ are wider than those observed for the other two cases. The width frequency at low temperatures changes between four or five orders of magnitude, and its intensity is lower than those for the SPEEK/Z8 and SPEEK/Z67 membranes. This observation suggests that the height and width of the peaks in tan δ are strongly related to the structure of the membranes. In our case, we have a decrease in the charge carriers when the amount of ZIFs increases due to the interaction between hydrogens from the sulfonic acid groups contained in the polymeric chains and nitrogen atoms from the organic linkers of ZIF compounds. These observations are in agreement with the observed IEC values. On the other hand, the hydrophobicity of ZIFs leads to the build up of aggregates, which get worse with the increase in charge transport. This effects cause a decrease of the amount of available charge carriers, which is reflected in a bigger distribution of relaxation times, producing a decrease in the peaks’ intensity and an increasing of the peak width. These effects are more evident when the interaction of the charge carriers grows with the increase of the ZIF loading. This phenomenon is very evident in the case of the samples containing ZIF-Mix at 10 wt. % at low temperatures.

When the concentration of ZIFs is above some critical value, we can see two peaks in the plot of Figure 8. It is due to the appearance of two different processes for the conductivity. Firstly, at low frequencies, some amount of dipoles cannot move with the electric field that is applied, due to the restrained mobility of the ions into the Debye length layer as a consequence of the opposite electric force created by the charges of the same sign. When the frequency increase appears, a second peak appears that is related to the DC conductivity of the membrane. This effect is dependent on the temperature producing an enhancement of the conductivity, and disappears with an increase in the temperature, since the electric interactions are broken; then only a peak in tan δ at high frequencies is observed, such as that which is observed from Figure 8.

In the present work, Equation (7) has been used to fit the experimental data shown in Figure 8 and obtain an estimation for the parameters M and τ_m_, for all of the temperatures. The values of these parameters are given in Table 1.

A closer inspection of Table 1 also permits observing that the relaxation time τ_m_ and the parameter M strongly depend on the temperature. The relaxation time (τ_m_) decreases with the temperature increase, whereas the parameter M increases. Figure 8 shows the fitting for three composite membranes—SPEEK/Z8, SPEEK/Z67, and SPEEK/ZMix—for a ZIF loading of 5 wt. % at different temperatures. The solid lines indicate the convolution of Equation (4) in the peak of tan δ at higher frequencies. As can be seen, curves display peaks corresponding to the maxima in tan δ, which are associated with the plateau of the real part of the conductivity observed in the Bode diagrams (Figure 7). Regardless of the model used, the value of the frequency of the peaks in tan δ corresponding to values in frequency are the same and related with the parameters M and τ_m_ as:(8)$$\omega_{\max}^{\tan\;\delta }=\frac{1}{\tau_{m}}$$

The M parameter can also be expressed as M = L/2L_D_, and the parameters τ_m_ and ε_EP_ are related to M through the expressions τ_m_ = M^1/2^·τ, with τ being the relaxation time, which is also defined as σ = ε_S_/σ_dc_, where ε_S_ is the dielectric permittivity of the sample [66,68]. The parameter L is the thickness of the sample (i.e., electrode separation when the sample is sandwiched to take the measurements), and L_D_ is the Debye length, which can be defined as:(9)$$L_{D}=\sqrt{\frac{\varepsilon_{S}\varepsilon_{0}\mathrm{kT}}{q^{2}n}}$$
where k is the Boltzmann constant, T is the absolute temperature, q is the ion charge, and *n* is the mobile charge density.

Considering that cations and anions have approximately the same mobility μ, the conductivity σ_dc_ can be expressed in terms of n and μ according to the following expression:(10)$$\sigma_{\mathrm{dc}}=\mathrm{nq}\mu$$
and the ion mobility can be determined as:(11)$$\mu =\frac{{\mathrm{qL}}^{2}}{{4M\tau }_{m}\mathrm{kT}}$$

Finally, considering the Einstein relation for ion diffusivity in combination with Equation (10) and Equation (11), the diffusion coefficient of the protons can be expressed as:(12)$$D=\frac{L^{2}}{{4M}^{2}\tau_{m}}$$

The use of Equation (12) allows calculating the ion diffusivity in terms of parameters such as M $\tau_{m}$ and L. From equations (10) and (12), the ionic charge density (n) is calculated as:(13)$$n=\frac{\sigma_{\mathrm{dc}}\mathrm{kT}}{{\mathrm{Dq}}^{2}}$$

The calculated values for diffusivity *D* and charge density *n* as a function of the temperature are shown in Figure 9. From Figure 9A, we can observe a similar effect in diffusivity *D* to that observed in conductivity. For SPEEK/Z8 and SPEEK/Z67 membranes, the diffusivity rises as the temperature increases; however, an abrupt decrease in the calculated diffusivity happens at 120 °C, which can be attributed to the dehydration of the membrane. Contrarily, for the SPEEK/ZMix membrane, the diffusivity decreases as the temperature increases.

Assuming the electrolyte is univalent, we have obtained the values of diffusion coefficients from the previously determined values of M and τ_m_, which are in agreement with Equation (12). The values that are found are reasonably similar to those compared with other systems, and decrease at higher temperatures.

Finally, the values of free-ion density have been determined through combining the values of DC-conductivity obtained from Bode diagrams and the values of diffusion coefficients calculated according to Equation (12). Our results showed that the calculated charge concentration density increases with temperature for the SPEEK/ZMix sample, which is practically constant for the SPEEK/ZIF8 sample and slightly decrease for the SPEEK/ZIF67 sample. Such results can be related to the different structures and compositions of the fillers, with the cross-linking and solvent. The results that were observed are in agreement with the behavior observed for the composites membranes of PEI/ZIFs [77,78]. From the Arrhenius fit of the free-ion density, we have estimated the dissociation energy. In general, for SPEEK/Z8 and SPEEK/Z67, the value of dissociation energy decreases when the concentration of ZIFs increases, yielding for SPEEK/Z8 1 wt. % and SPEEK/Z8 10 wt. %, 0.80 ± 0.01 eV and 0.12 ± 0.01 eV and 0.67 ± 0.08 eV and 0.06 ± 0.02 eV for SPEEK/Z67 at 1 wt. % and 10 wt. % loading, respectively. The tendency is different for SPEEK/ZMix that has the highest value for 10% concentration of ZIF with a dissociation energy of 0.70 ± 0.14 eV. When we compared the composite membranes at one concentration, the tendency is SPEEK/Z8 > SPEEK/Z67 > SPEEK/ZMix at lower concentrations, and shows an opposite behavior at higher concentrations.

## 4. Conclusions

In summary, we have prepared and characterized polymeric composite membranes based on SPEEK containing three MOFs (ZIF-8, ZIF-67, and ZIF-Mix) as fillers at different loadings by means of water uptake, swelling ratio, IEC, TGA, and EIS. The charge transport mechanism has been studied through an electrode polarization model applied over experimental data, according to the methodology followed by Bandara et al., which fit the tan δ = ε″/ε′ versus frequency. From the analysis of results, we have found that a bit overestimates free ion diffusivity, while underestimating the free-ion number density, although the values are in the same range as those reported in other systems studied with a Trukhan electrode polarization model.

Proton conductivity showed a dependence on the ZIFs content, reaching values as high as 29 mS·cm^−1^ at 100 °C for a SPEEK/ZMix at 1 wt. %. Interesting values of conductivity were observed for all of the samples. These conductivity values indicate that these composite membranes represent a promising alternative to Nafion for its application in different energy devices. These results, in combination with their thermal stability and good physicochemical properties, make them potential candidates to operate as IT-PEMFCs.

## Figures and Tables

**Figure 1 nanomaterials-08-01042-f001:**
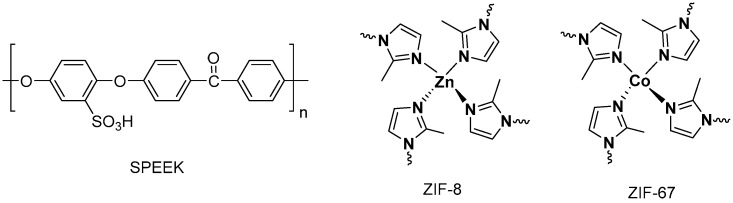
Chemical structure of sulfonated poly(ether-ether-ketone) (SPEEK) polymer repeating unit and zeolitic imidazolate framework (ZIF)-8 and ZIF-67.

**Figure 2 nanomaterials-08-01042-f002:**
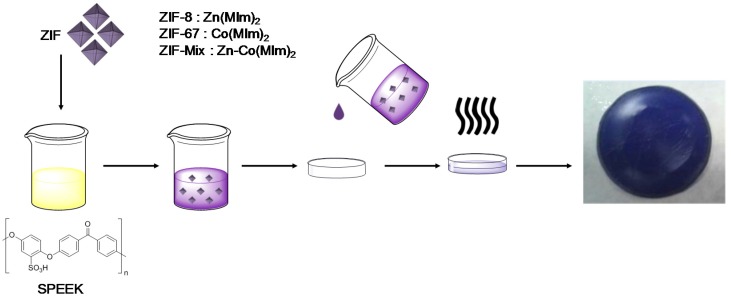
Schematic representation of casting method for membrane preparation.

**Figure 3 nanomaterials-08-01042-f003:**
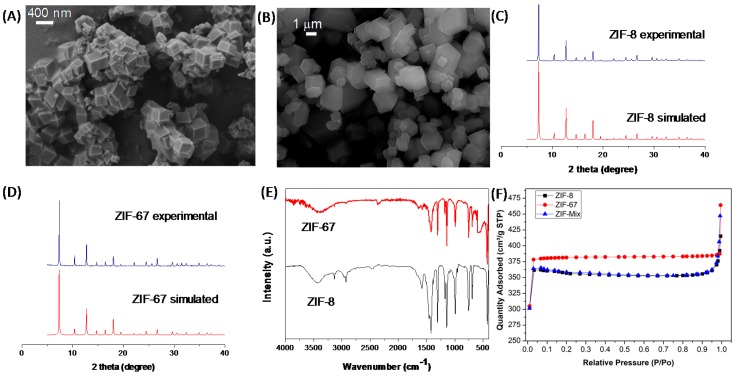
Field emission scanning electron microscopy (FE-SEM) images of (**A**) ZIF-8 and (**B**) ZIF-67. (**C**) XRD patterns of simulated ZIF-8 (red) and ZIF-8 as synthesized (blue). (**D**) XRD patterns of simulated ZIF-67 (red) and ZIF-67 as synthesized (blue). (**E**) FT-IR spectra of ZIF-8 and ZIF-67. (**E**) N_2_ adsorption-desorption isotherms at 77 K for ZIF-8, ZIF-67 and ZIF-Mix.

**Figure 4 nanomaterials-08-01042-f004:**
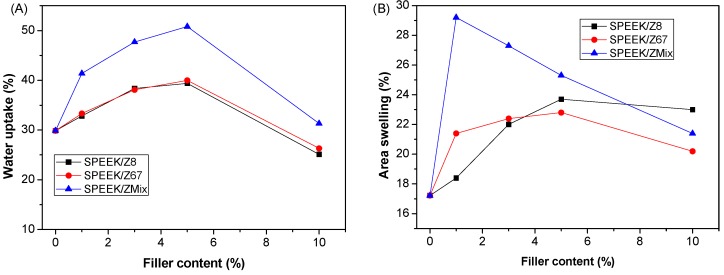
(**A**) Water uptake and (**B**) swelling degree of the pristine SPEEK membrane and ZIF-composite membranes with different filler content (wt. %).

**Figure 5 nanomaterials-08-01042-f005:**
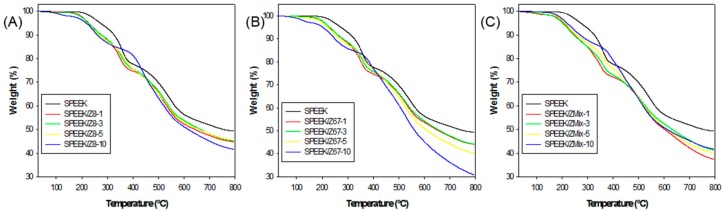
Thermogravimetric analysis (TGA) of the ZIF-composite membranes with different filler content (wt. %): (**A**) SPEEK/Z8, (**B**) SPEEK/Z67 and (**C**) SPEEK/ZMix.

**Figure 6 nanomaterials-08-01042-f006:**
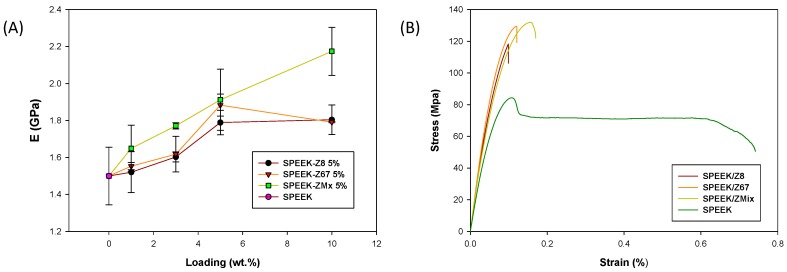
(**A**) Young modulus and (**B**) stress–strain curves for pure SPEEK membrane and composite membranes containing 5 wt. % of ZIFs.

**Figure 7 nanomaterials-08-01042-f007:**
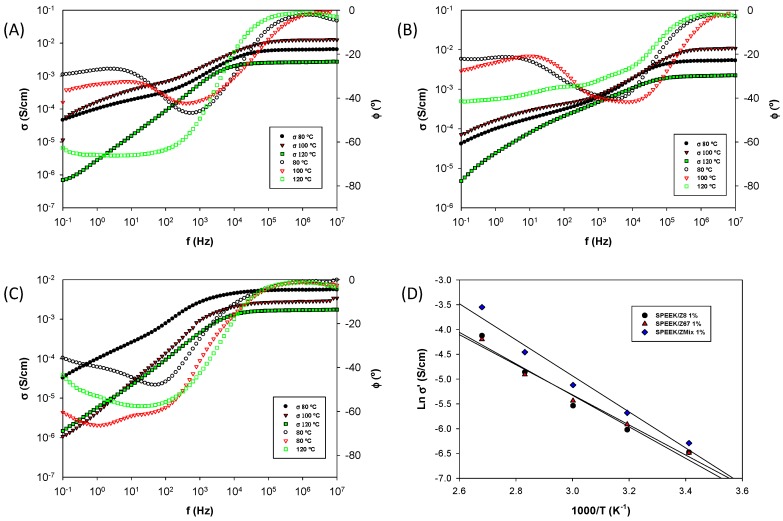
Bode diagrams for the samples: (**A**) SPEEK/Z8, (**B**) SPEEK/Z67, and (**C**) SPEEK/ZMix 3% at temperatures of 80 °C, 100 °C, and 120 °C, respectively. (**D**) Arrenhius plot of the SPEEK/ZIFs membranes for 3 wt. % of ZIFs concentration.

**Figure 8 nanomaterials-08-01042-f008:**
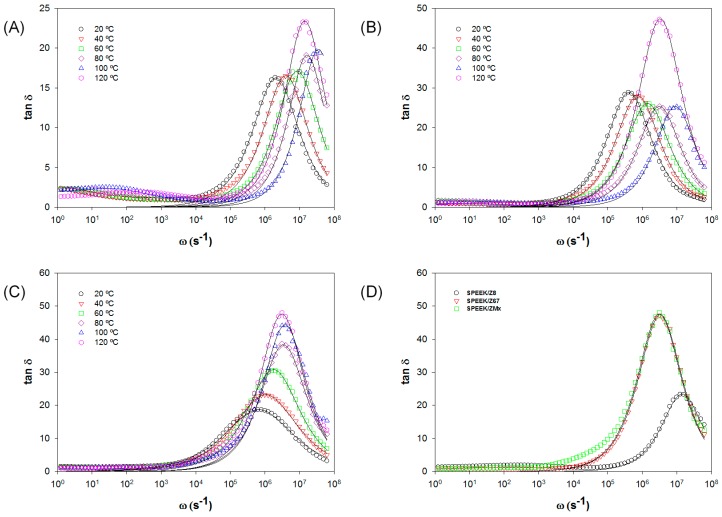
Tan δ vs. frequency in all the range of temperatures for 5% loading (**A**) SPEEK/Z8, (**B**) SPEEK/Z67, (**C**) SPEEK/ZMix, and (**D**) the three different ZIFs studied at 120 °C at 5 wt. % loading.

**Figure 9 nanomaterials-08-01042-f009:**
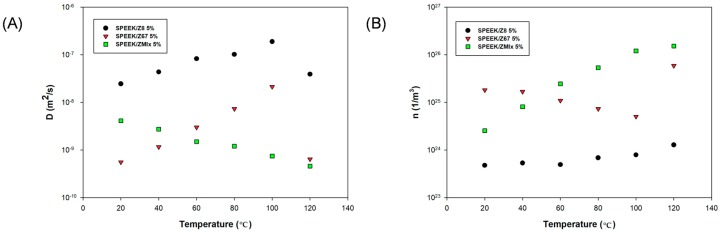
(**A**) Plot of diffusivity vs. temperature and (**B**) plot of charge density vs. temperature for SPEEK composite membranes with a ZIF loading of 5 wt. %.

**Table 1 nanomaterials-08-01042-t001:** Calculated parameters M and τ_m_ obtained from fitting experimental data of tan δ for the composite membranes SPEEK/Z8, SPEEK/Z67, and SPEEK/ZMix at 1 wt. % filler loading.

Membrane	T = 40 °C			T = 80 °C			T = 120 °C		
	M	τ_m_ × 10^7^ (s)	D × 10^7^ (cm^2^/s)	M	τ_m_ × 10^7^ (s)	D × 10^7^ (cm^2^/s)	M	τ_m_ × 10^7^ (s)	D × 10^7^ (cm^2^/s)
SPEEK/Z8	790	2.00	80.1	12,580	0.35	1.81	19,000	1.30	0.21
SPEEK/Z67	990	3.0	76.5	3420	1.00	19.2	5200	2.50	3.33
SPEEK/ZMix	1580	7.00	5.16	11,460	0.8	0.6	18,800	1.18	0.22

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
