# Peer review of "Enhanced Conductivity of Composite Membranes Based on Sulfonated Poly(Ether Ether Ketone) (SPEEK) with Zeolitic Imidazolate Frameworks (ZIFs)"

_nanomaterials, 2018, doi:10.3390/nano8121042_

Reviewer 1 Report

The manuscript entitled "Conductivity study of composite membranes based sulfonated poly-ether-ketone (SPEEK) with different concentrations of zeolitic imidazolate frameworks (ZIFs) for proton conductivity applications" reports about the formulation and characterization of SPEEK-based membranes, potentially suitable for applications in electrochemical devices.

In my opinion, it is a good work: the experimental procedures are well designed and the obtained results are well presented.

Therefore, I suggest the publication of the submitted manuscript on Nanomaterials after that the authors have adressed the following issues:

In my view, the title of the paper should be shortened, in order to make it more appealing for potential readers.

In the experimental part, I propose to move the paragraph 2.3 before of "Characterization"

I suggest to the authors to move Figure 3  and the related comment in the experimental part, after the paragraph "Membrane preparation"

Author Response

Reviewer 1

The manuscript entitled "Conductivity study of composite membranes based sulfonated poly-ether-ketone (SPEEK) with different concentrations of zeolitic imidazolate frameworks (ZIFs) for proton conductivity applications" reports about the formulation and characterization of SPEEK-based membranes, potentially suitable for applications in electrochemical devices.

In my opinion, it is a good work: the experimental procedures are well designed and the obtained results are well presented.

Therefore, I suggest the publication of the submitted manuscript on Nanomaterials after that the authors have adressed the following issues:

In my view, the title of the paper should be shortened, in order to make it more appealing for potential readers.

We thank reviewer 1 for his/her comments regarding our manuscript. According to his comment, we have shortened the title of the paper: “Enhanced conductivity of composite membranes based on sulfonated poly-ether-ether-ketone (SPEEK) with zeolitic imidazolate frameworks (ZIFs)”.

"In the experimental part, I propose to move the paragraph 2.3 before of "Characterization"

We agree with reviewer 1 proposal and therefore, we have moved the paragraph 2.3 entitled Experimental procedures. Consequently, these sections have been renumbered and now sections 2.2. and 2.3. correspond to Experimental procedures and Characterization, respectively.

I suggest to the authors to move Figure 3  and the related comment in the experimental part, after the paragraph "Membrane preparation"

We agree with reviewer 1 comment and we have moved Figure 3 and the related comment located in Results and disussion, after the paragraph "Membrane preparation".

Yours sincerely

Vicente Compañ

Reviewer 2 Report

Very nice and interesting work the entitled: "Conductivity study of composite membranes based sulfonated poly-ether-ether-ketone (SPEEK) with different concentrations of zeolitic imidazolate frameworks (ZIFs) for proton conducting applications"

I agree for the acceptance of this publication after some minor clarifications:

- The authors claim that the composite membranes can be used at applications above 100 oC but since their thermal stability is worse with the incorporation of the MOF particles what is the final applied temperature range for these type of membranes?

- Are there references for practical applications of the composites?

- What is the benefit of using the prepared mixed matrix membranes instead of the pure membranes?

Author Response

Reviewer 2

Very nice and interesting work the entitled: "Conductivity study of composite membranes based sulfonated poly-ether-ether-ketone (SPEEK) with different concentrations of zeolitic imidazolate frameworks (ZIFs) for proton conducting applications"

I agree for the acceptance of this publication after some minor clarifications:

We thank reviewer 2 for his/her kind comments regarding our manuscript.

- The authors claim that the composite membranes can be used at applications above 100 °C but since their thermal stability is worse with the incorporation of the MOF particles what is the final applied temperature range for these type of membranes?

We thank reviewer 2 for this comment and we agree that the thermal stability of the composite SPEEK membranes containing ZIFs as fillers is worsened in comparison to the pure SPEEK membrane for high temperatures. However, the final temperature range for the practical application of these type of membranes is around 100-140 °C. Therefore, this worsening would not affect the membranes operating range. As we conclude from our results, the proton conductivity of our composite membranes showed a dependence on the ZIF content, reaching values as high as 0.029 S/cm at 100 °C for SPEEK/ZMix at 1% wt. Such values of conductivity indicate that these composite membranes represent a promising alternative to Nafion for its application in different energy devices, because Nafion conductivity value at 100 ºC is similar to our result in SPEEK-ZIFs membranes.  However, the Tg of Nafion membranes is around 105 ºC below than SPEEK membranes where Tg is around 180 ºC  making them candidates to be used above 100 °C, which would substantially improve the efficiency and cost of the system as a PEM to be used as Membrane electrode assembly in a Fuel Cell.

These results in combination of its thermal stability and good physicochemical properties make them potential candidates to operate as PEMFCs in the temperature range of 100 to 130ºC.

We have rewritten the final part of the thermal stability discussion and it states like: “It should be noted that although SPEEK composite membranes showed a slightly lower thermal stability than pure SPEEK membrane, there is not a significant worsening in the in the operational temperature range (100–140 °C). Therefore, we can conclude that composite membranes possess high thermal stability in the operational range for IT–PEMFC membranes.”

- Are there references for practical applications of the composites?

As can see in the introduction, we explicitly say that: “the incorporation of MOFs into polymeric SPEEK membranes has shown a significant improvement of the performance in fuel cell applications]. Among the reported MOF-containing PEMs, proton conductivities of 268 and 306 mS·cm−1 have been reported for SPEEK membranes containing sulfonated UiO–66 and MIL–101, respectively with under 100% RH and temperatures below 70 °C. However, these values decrease at temperatures higher than 100 °C and anhydrous or low humidity conditions. One family of MOFs are zeolitic imidazolate frameworks (ZIFs), which are neutral porous framework structures with high chemical and thermal stability based on imidazolate rings coordinated to a tetrahedral divalent metal cation. The use of this subclass of MOFs has also been demonstrated in the preparation of PEMs, reaching proton conductivities up to 24 and 50 mS·cm–1 at 120 °C and 30% RH for SPEEK membranes with ZIF-8 and ZIF-8/CNT hybrid cross-linked networks, respectively [46].”

All the references included in this approach show that composite membranes similar to our composites have been studied for energy device applications as PEMFC and DMFC. In our work, we probe that conductivities of SPEEK/ZIFs composite membrane reach values of conductivity close to 30 mS/cm at 100 ºC. On the other hand, studies of DSC for SPEEK membranes showed a Tg around 180ºC. Such results clearly indicate that our composite membranes can be used as polyelectrolyte in a PEMFC and DMFC. The results found in this paper put of manifest the possibility to use our composite for PEMFC at 120ºC. The application of these composite membranes as single cell performances for fuel cell applications of the SPEEK membranes is being currently studied and our intention is to publish the results in a future communication.In this paper, studies of performance in a single PEMFC will show a very  good performance for H2/O2 fuel cell at 100 and 120ºC.

- What is the benefit of using the prepared mixed matrix membranes instead of the pure membranes?

We thank reviewer 2 for his/her comment. As mentioned in the Results and discussion section, the addition of ZIF compounds improves the proton conductivity when compared with the pure SPEEK membrane. Additionally, yater uptake and Young modulus increase upon incorporation of the ZIF as filler. As consequence of the increasing of water uptake, the conductivity at temperatures between 100 and 120 ºC increases.

 Yours sincerely

Vicente Compañ